# The Effectiveness of Pilates Training Interventions on Older Adults’ Balance: A Systematic Review and Meta-Analysis of Randomized Controlled Trials

**DOI:** 10.3390/healthcare11233083

**Published:** 2023-12-01

**Authors:** Tatiana Sampaio, Samuel Encarnação, Olga Santos, Diogo Narciso, João P. Oliveira, José E. Teixeira, Pedro Forte, Jorge E. Morais, Catarina Vasques, António Miguel Monteiro

**Affiliations:** 1Department of Sports Sciences, Instituto Politécnico de Bragança, 5300-253 Bragança, Portugal; tatiana_sampaio30@hotmail.com (T.S.); samuel01.encarnacao@gmail.com (S.E.); a38511@alunos.ipb.pt (O.S.); a41462@alunos.ipb.pt (D.N.); jose.eduardo@ipb.pt (J.E.T.); morais.jorgestrela@gmail.com (J.E.M.); catarinav@ipb.pt (C.V.); mmonteiro@ipb.pt (A.M.M.); 2Research Center in Sports Sciences, Health Sciences & Human Development (CIDESD), 5001-801 Vila Real, Portugal; joao.pedro.costa.oliveira@ubi.pt; 3Department of Physical Activity and Sport Sciences, Universidad Autónoma de Madrid (UAM), Ciudad Universitaria de Cantoblanco, 28049 Madrid, Spain; 4Department of Sports Sciences, University of Beira Interior, 6201-001 Covilhã, Portugal; 5Department of Sports Sciences, Polithecnic Institute of Guarda, 6300-559 Guarda, Portugal; 6CI-ISE, Instituto Superior de Ciências Educativas do Douro, 4560-547 Penafiel, Portugal; 7Research Centre in Basic Education (CIEB), Instituto Politécnico de Bragança, 5300-253 Bragança, Portugal

**Keywords:** physical exercise, prevention, falls, mortality, geriatric population, postural balance

## Abstract

Background and Objectives: Pilates training intervention programs have gained attention as a potential approach to enhancing balance in older adults, thereby reducing the risk of falls. In light of these considerations, this systematic review and meta-analysis aimed to critically evaluate the existing evidence and determine the effect of Pilates training intervention programs on older adults’ balance. Materials and Methods: The literature was searched through the PubMed, Web of Science, and Scopus databases from inception until July 2023. The primary keywords used for the literature search included “elderly” or “older adults” and “pilates training” and “balance”. Results: The systematic review through qualitative analysis showed robust evidence about the efficacy of Pilates intervention programs in improving older adults’ balance. The pooled meta-analysis of static and dynamic balance showed that eight (53%) out of a total fifteen analyzed interventions presented a significant effect of Pilates in improving the participants’ balance, without between-study heterogeneity. In addition, the meta-analysis regarding dynamic balance showed that six (67%) out of nine analyzed interventions presented a significant effect of Pilates in improving the participants’ balance, without heterogeneity between studies. Similarly, the meta-analysis regarding static balance showed that four (50%) out of eight analyzed studies presented significant effects on the older adults’ balance, where moderate between-study heterogeneity was found. Sensitivity analysis showed that three studies reduced the between-study heterogeneity (19, 17.6, and 17%), regressing from moderate to low heterogeneity, *p* < 0.05. Conclusions: This systematic review and meta-analysis underscores the potential of Pilates training as a valuable intervention to enhance balance in the elderly population.

## 1. Introduction

Globally, the number of people over 60 years is increasing at an annual rate of 3%, much higher than in the younger age groups. Forecasts indicate that by 2050, the elderly will account for 22% of the population [1]. One of the most common causes of falls among the elderly is believed to be stumbling into obstacles [2]. In addition to slower crossing speeds, shorter step lengths, and shorter heel–obstacle distances compared to young people, it was found that older individuals increased the distance between the lower limbs (i.e., inter-limb distance/space between the feet) [3]. 

A factor that significantly contributes to falls in older individuals is the increased heel–obstacle distance to avoid stumbling, which can cause imbalance disorders and changes in body posture [4]. A frontal-pelvic movement [4], greater variability in the interarticular coordination of the lower limbs [5], and an altered center of mass movement pattern [6], among other changes in the whole body posture, may increase the risk of imbalance during the transposition of obstacles. The inability to regain balance after stumbling or loss of balance can result in falls. It is therefore crucial to understand the changes in the ability of the elderly to maintain balance before designing and evaluating strategies to reduce the risk of falling. In the realm of fall prevention and balance enhancement for older individuals, various training programs are in place, with multi-factor exercises often proving more effective [7]. While some programs focus on specific aspects like strength and endurance training [8], the Pilates method, invented by Joseph Pilates in 1920, has gained attention for its comprehensive approach. Exercises that combine the body and mind are used and require concentration in breathing and body posture to keep the trunk’s stability, and thus improve the muscle strength, body control, and flexibility of the practitioners [9]. Six key elements are used: center, concentration, control, precision, fluidity, and breathing. The principle of “Center” underscores the engagement of core musculature to establish and maintain stability. “Concentration” is a foundational aspect, necessitating focused cognitive attention to execute each movement deliberately. The principle of “Control” permeates the methodology, demanding precision in the execution of exercises. “Precision” itself is a guiding tenet, mandating meticulous attention to form and alignment. The concept of “Fluidity” is embraced, promoting seamless and connected movements throughout the entirety of the practice. Lastly, “Breathing” assumes a specific physiological role, intricately synchronized with each exercise to optimize respiratory function and enhance mind–body integration [9]. It can be performed alone or in groups, with equipment (such as the reformer), or on the floor (with a carpet), or only with the body weight [10]. The effectiveness of the Pilates technique is thus revealed, which provides psychomotor advantages, increases functional capacity, promotes independence, and increases the quality of life [11].

In recent years, there has been a gradual increase in studies examining the efficacy of Pilates training [12]. Furthermore, research has strongly supported the effectiveness of Pilates in various health, physiotherapy, and rehabilitation contexts, offering promising insights into its multifaceted applications, particularly in post-injury or post-surgery rehabilitation [13]. Pilates has proven to be a valuable asset in addressing a spectrum of issues, including musculoskeletal injuries, back pain, and joint problems, demonstrating its versatility and adaptability in aiding individuals on the path to recovery [14]. 

A notable research gap remains despite the growing interest in Pilates training interventions aiming to enhance balance in older adults. A systematic review conducted in 2018 by Moreno-Segura et al. [12] endeavored to explore this subject; however, it was constrained by its inclusion of only a limited number of studies, consequently yielding a restricted overview of the existing empirical evidence. In the meantime, a considerable body of experimental research has emerged, warranting a comprehensive reassessment of the literature to establish a more robust comprehension about Pilates training’s effects on older adults’ balance. This prevailing research gap underscores the imperative for an updated and exhaustive systematic review and meta-analysis that can encompass a more extensive array of studies and thus provide a more comprehensive, evidence-based perspective on this critical subject matter. In light of these considerations, this systematic review and meta-analysis aims to assess the existing evidence critically and ascertain the impact of Pilates training interventions on the balance, both static and dynamic properties, of older adults.

## 2. Materials and Methods

### 2.1. Literature Search and Article Selection

This systematic review was carried out following the recommendations and criteria outlined in the Preferred Reporting Items for Systematic Reviews and Meta-Analysis Statement (PRISMA) [15]. To update information related to the effect of Pilates training programs on older adults’ balance, articles published by 2023 have been included. The study protocol was registered in PROSPERO under the code CRD42023469540.

A systematic review of the literature was conducted by a researcher (TS) to identify and evaluate articles on Pilates training in the elderly. Three databases, comprising PubMed Medline, Web of Science (Web of ScienceSM; Current Contents Connect), and Scopus, were in use from the start until 20 May 2023. The keywords used for the research were “elderly” or “older adults”, “Pilates training”, and “balance”. The references to all the retrieved studies were reviewed to detect other potentially relevant studies not identified by surveys in the databases.

Two authors independently reviewed the titles and summaries of the identified articles. In cases of doubt as to the eligibility of an article, the full text has been collected. Two independent authors (D.N. and O.S.) reviewed the articles in the selection and evaluated the eligibility requirements. Two authors (D.N. and O.S.) independently evaluated each article in two phases of sorting: the title, the summary, and, subsequently, the full text of the article. Conflicts over eligibility have been resolved through dialogue and, if necessary, with the assistance of a third author (T.S.).

### 2.2. Inclusion and Exclusion Criteria

The eligibility criteria have been structured using the PICOS framework (P: population, I: intervention or independent variable, C: comparators, O: outcomes, and S: study design). The population (P) studied consisted of healthy older adults (≥60 years) of both genders. The intervention (I) eligible studies used Pilates training as an intervention method, with a minimum intervention period of four weeks. Pilates training could be performed on a carpet, a reformer, or both. Comparators (C) were defined as studies with a control group for comparison. Therefore, the outcomes (O) analyzed were those that assessed balance as either static or dynamic. Finally, the included studies (S) were only randomized controlled trials (RCTs). Summaries of lectures and review articles were excluded.

The exclusion criteria included studies that implemented other training methods, articles without full-text availability, and studies published in languages other than English. Additionally, gray literature, webpages, and Google Scholar were not considered.

### 2.3. Data Extraction

Two authors (T.S. and O.S.) independently extracted the characteristics and results of the interventions in each included publication, according to the PRISMA statement [15]. The first author, year of publication, information on the characteristics of the participants (sample size, sex of the participant, average age, and standard deviation), duration of the intervention, frequency of the training program, and the instrument used to assess balance, as well as the main results, were extracted from each study.

### 2.4. Risk of Bias and Quality Evaluation of Studies’ Quality

The studies’ quality and risk of bias were assessed using a 14-item scale for the Quality Assessment Tool for Controlled Intervention Studies based on the National Institutes of Health’s (NIH) recommendations, which focus on the main concepts for evaluating a study’s internal validity [16], as shown in Appendix A.

### 2.5. Synthesis Methods

When identifying studies that measured the effects of Pilates training on older adults’ balance, following the Cochrane Handbook Guidelines [17], we converted mean differences into Cohen’s d effect sizes and their respective standard errors, which enabled us to perform a meta-analysis of effect sizes. Then, we summarized the pooled and isolated effect sizes and standard errors provided by the comparison between experimental and control groups in forest plots. With this aim, we performed a meta-analysis of the obtained effect sizes. As the principal metanalysis’s outputs, we used the Cochran Q-test to measure heterogeneity and *I*^2^ statistics to measure individual studies’ inconsistency, ranging from 0 (any inconsistency) to 100% (maximal inconsistency) [18]. We measured the τ^2^ to verify the between-study variance. When we verified the methodological heterogeneity between the included studies, we applied a random effects metanalysis; if heterogeneity was not verified, we applied a fixed-effects meta-analysis [18]. We performed three meta-analyses, considering pooled data (inserting static and dynamic balance effect sizes), effect sizes for dynamic balance, and effect sizes for static balance [19]. Both meta-analyses reported the 95% confidence intervals (ICs) for the effect size of Pilates training on older adults’ balance. When meta-analysis presented K ≥ 10 studies, we performed a funnel plot analysis to measure the study’s publication bias [18]. In addition, we performed a sensitivity analysis using the leave-one-out method to verify the individual study’s influences on the pooled meta-analyses’ effects [18]. All analyses were performed in R, the statistical programming language, Version 4.3.1 [20].

## 3. Results

### 3.1. Study Selection

Through the search in the PubMed/Medline, Web of Science, and Scopus databases, we found 348 records. We removed 118 duplicates and excluded 204 after evaluating the title and abstract. Subsequently, we organized these studies into distinct categories, including those with inappropriate study designs, participants below the age of 60, studies not employing objective measures to assess balance and those with outcomes that did not align with our criteria. Thus, we assessed the 26 remaining studies by reading the relevant sections. We excluded six studies that did not meet the inclusion criteria, grouping them into insufficient intervention duration (*n* = 4) and non-compliance with Pilates training (*n* = 2). Finally, 20 studies that met the criteria and aims of this systematic review were included, as shown in Figure 1.

### 3.2. Study Characteristics

The 20 studies examined had a total of 930 individuals, with a mean age ranging from 63 to 79. With a total sample size ranging from 27 participants in the study by Bird et al. [21] to 107 individuals in the study by Aibar-Almazán et al. [22], all studies had more than 25 participants. Older individuals were the test subjects, and the intervention duration ranged from 4 weeks to 18 weeks in the studies from Mesquita et al. [23] and Carrasco-Poyatos et al. [24], respectively.

Table 1 below summarizes key details from the included studies for the meta-analysis, focusing on the characteristics of the study populations and the Pilates intervention parameters. These details include the first author’s name/publication year, the sample size, age, and sex distribution of the participants, as well as information on the volume, frequency, and duration of the Pilates intervention employed in each study. In cases where the sample size was not reported for the intervention group or the control group, the total number of participants was reported. The same was applied to the age of the participants.

### 3.3. Quality and Risk of Bias of Individual Studies

We evaluated all of the included studies in the systematic review following the guidelines of the NIH Quality Assessment Tool for Controlled Intervention Studies [16].

The following studies presented a minimal risk of bias, and their quality was considered good [21,22,23,26,27,28,29,30,31,33,34,36,37,38,39]. However, in some studies, a risk of bias was identified, and their quality was considered poor [24,25,32,40]. One study was considered to be of fair quality [35]. The studies did not meet some of the items, and the common flaws included the blinding of participants and providers, blinding of outcome assessors, lack of adherence to intervention protocols, uniformity of interventions, and sample size adequacy. We detailed this information in Table 2.

### 3.4. Results of Individual Studies

#### 3.4.1. Correlation between Pilates Training and Static Balance

Objectively, the static balance was measured in 11 studies [21,22,23,24,25,27,28,29,30,35,37]. The studies’ publication year ranged from 2012 in the study of Bird et al. [21] to 2022 in the study of Da Silva et al. [29]. In the evaluation of static balance, the majority of studies employed the force platform as the measurement instrument [21,22,23,24,25,27,30,33,37,39], with one study utilizing a barometric platform [28] and another using the Tinetti scale [35].

For the Pilates training interventions’ impact on balance in older adults, we analyzed the findings from various studies utilizing different measurement instruments. The mean difference ranged from MD = −2.59 in the study of Roller et al. [34] to MD = 6.34 in the study of Barker et al. [25]. Table 3 provides a summary of key details of the studies that measured static balance, focusing on the characteristics of the study, including the instrument used and the main results.

#### 3.4.2. Correlation between Pilates Training and Dynamic Balance

The dynamic balance was objectively measured in 16 studies [21,23,25,27,28,29,30,31,32,33,34,36,37,38,39,40]. The studies’ publication year ranged from 2012 in the study of Bird et al. [21] to 2022 in the study of Da Silva et al. [29]. In the evaluation of dynamic balance, the majority of studies employed the TUG test as the measurement instrument [21,23,25,26,27,28,29,32,33,34], four studies utilized a Berg balance scale [23,26,30,36], one wielded the force platform [37], one used the dynamic gait index [25], one used the four square step test [21], one utilized the dynamic stability measurement platform [31], one used the Fullerton advanced balance scale [32], one utilized the functional reach test [23], and one used the Y-balance test [39].

The mean differences ranged from MD = −3.60 in the study of Mesquita et al. [23] to MD = 2.49 in the study of Da Silva et al. [29]. Nevertheless, only one study by Bird et al. [21] reported an effect size of d = −0.50 (CI = −1.51 to 0.50), *p* < 0.467. Table 4 provides a summary of key details of the studies that measured dynamic balance, focusing on the characteristics of the study, including the instrument used and the main results.

### 3.5. Meta-Analysis Results

In the course of the systematic review, 20 studies were identified that met the criteria and aims outlined (as indicated in Figure 1). However, it is essential to note that, despite meeting the inclusion criteria, not all studies were included in the subsequent meta-analyses presented in Table 1. This discrepancy arose from methodological variations among the included studies, particularly in the diverse range of instruments employed for assessing balance outcomes. Therefore, to maintain methodological homogeneity and ensure the reliability of the pooled results, a selective approach was adopted for the meta-analyses. A pooled meta-analysis that considered the dynamic and static balance effect sizes showed that eight studies (53%) [21,23,25,26,28,29,33,35] of the 13 analyzed studies presented significant and positive effects observed individually of Pilates training on the older adults’ dynamic and static balance. The overall estimated effect size was estimated at 0.65, SE = 0.0817, Z = 7.985, *p* < 0.001, CI [0.4924 to 0.8127]. These results were obtained from a fixed-effects meta-analysis, since we did not find significant between-study heterogeneity Q = 19.0455, df = 14, *p* = 0.1632, *I*^2^ = 26.5%, as presented in Figure 2.

Regarding dynamic balance, the meta-analysis showed that six (67%) [21,25,26,28,29,33] out of nine studies presented isolated significant effects in favor of Pilates training in older adults’ dynamic balance. The overall meta-analysis confirmed this significance: estimate = 0.7334, SE = 0.1153, Z = 6.3612, *p* < 0001, CI [0.5074 to 0.9594]. These results were provided from a fixed-effects meta-analysis, since we did not verify significant between-study heterogeneity: Q = 8.6256, df = 8, *p* = 0.3749, *I*^2^ = 7.25%, as presented in Figure 3.

Regarding static balance, the meta-analysis showed that four (50%) [23,25,26,35] out of eight included studies presented isolated significant effects in favor of Pilates training in older adults’ static balance. The overall meta-analysis confirmed this significance: estimate = 0.5570, SE = 0.1512, Z = 3.6837, *p* = 0.0002, CI [0.2606 to 0.8534]. These results were provided from a random-effects meta-analysis, since we verified a trend towards significant between-study heterogeneity: Q = 13.8153, df = 7, *p* = 0.05, *I*^2^ = 49.6%, as presented in Figure 4.

#### Sensitivity Analysis Results

We did not perform the funnel plot analysis to verify the between-study asymmetry due to the unique meta-analysis that presented a trend for between-study heterogeneity (see Figure 4), which analyzed only eight studies; this broke the assumption about at least 10 prior analyzed studies to perform this type of analysis. In this way, to test the meta-analysis results, we verified the individual study’s influence through a sensitivity analysis based on the cut-offs of 0–24% = low, 25–49% = moderate, and 50–70% = high heterogeneity across studies, as suggested by Higgins et al. [18].

Regarding the pooled meta-analysis’s effects, we find that the studies of Carrasco-Poyatos et al. [27] (*I*^2^ = 19.6%), Irez et al. [31] (*I*^2^ = 17.6%), and Roller et al. [34] (*I*^2^ = 17%) reduced the between-study heterogeneity from a moderate to a low score. Nevertheless, this variability was not statistically significant, *p* > 0.05. After this analysis, the effect sizes ranged from 0.62 to 0.70 (Table 5).

Regarding the meta-analysis’s effects on dynamic balance, we did not find any individual study influence on the pooled results (*p* > 0.05). In addition, the between-study heterogeneity remained low (<24%) for all studies (Table 6).

Regarding the meta-analysis’s effects on static balance, depicted in Table 7, we found that the studies of Oliveira et al. [26], (*p* = 0.04, *I*^2^ = 54.6%) and Gabizon et al. [30], (*p* = 0.04, *I*^2^ = 53%) significantly influenced the pooled meta-analysis’s effects. In addition, the overall estimated effect sizes ranged from 12 points (0.49 to 0.62).

## 4. Discussion

The aim of this systematic review was to critically assess the existing evidence and ascertain the impact of Pilates training interventions on the balance, both static and dynamic, of older adults. We proved our main hypothesis when both static balance and dynamic balance presented significant improvements in the majority of the Pilates interventions.

Regarding the pooled meta-analysis considering static and dynamic balance, eight of the thirteen studies included in the meta-analysis revealed significant and positive effects of the Pilates training interventions. These results align with previous research findings, as a prior systematic review conducted by Bullo et al. [41] observed that Pilates interventions yield various favorable outcomes, including improvements in balance, increased functional mobility, and enhanced postural stability, consequently contributing to a reduced risk of falling. 

In the context of older adults, previous studies have identified several risk factors for falls, encompassing physiological decline, chronic diseases, psychological factors, and medication usage [42]. Among these contributing factors, balance assumes a central role, relying on sensory systems such as vision, vestibular sensation, and proprioception, as well as the central nervous system and skeletal muscles. Balance and gait issues have been identified as primary contributors to falls in older adults [43]. 

Notably, vestibular dysfunction is a significant risk factor for falls in this population and impairs the perception of body posture, movement, orientation, and motion. This dysfunction can disrupt body balance, amplifying the risk of falls and frequently resulting in dizziness [43]. In the study from [44], the authors found that the key risk factors for new falls in older persons were the deterioration of balance sensory input function (primarily vestibular and visual sense), skeletal muscle motor function, and related postural control capacity. Under these circumstances, Pilates training interventions offer many benefits, encompassing improvements in both static and dynamic balance and other health factors, thereby enhancing the overall well-being of older adults.

The majority of the studies included in our review seemed to support this conclusion. These results suggest that Pilates is a useful kind of exercise for enhancing balance in older persons, which is known to lower the risk of falls [45]. If the exercise guidelines for fall prevention are followed, more significant effects may be achieved. Also, exercises that test your ability to maintain your balance include those that are multisensory (for example, performed on various surfaces like mats, wobble boards, and discs, etc.) and are performed standing up; use a foot position that achieves a narrow base of support (for example, standing on one leg like in a scooter or standing leg pump exercise); and do not require the use of hands for support [46].

Concerning static balance, four of the eight studies included in the meta-analysis revealed significant and positive effects of the Pilates training interventions. In a pilot study conducted by Bergamin et al. [47], it was reported that Pilates yielded improvements in static balance, specifically mediolateral oscillations with open eyes, in postmenopausal women aged 59–66 years old. These findings align with the overall trend observed in our meta-analysis, which underscores the efficacy of Pilates in enhancing static balance among older adults. The pivotal role of static balance in fall prevention, closely linked to the vestibular system, underscores the significance of interventions aimed at enhancing this aspect, such as Pilates training, are crucial. Numerous studies [22,28,29,37,46], have consistently demonstrated the effectiveness of Pilates in improving various static balance parameters. This improvement not only contributes to a reduced risk of falls in older adults but also underscores the value of Pilates as a valuable intervention to enhance overall well-being and independence in this population [41].

In line with the preceding findings, dynamic balance also exhibited notable and positive improvements in six out of the nine studies examined. Dynamic balance pertains to maintaining equilibrium while in motion and sustaining or recovering a specific posture by ensuring the center of gravity remains within the base of support, a crucial skill to prevent stumbling or falls [48]. A study by Bergamin et al. [47], which incorporated a twice-a-week, 60 min Pilates exercise intervention over a 12-week period, highlighted that the Pilates method offers a valuable means to enhance one’s capacity for maintaining bodily balance. These positive effects likely stem from improved functionality in postural muscles and enhanced the integration of information from diverse sensory receptors, ultimately contributing to better control of these muscles.

Similar to our study, in another previous systematic review [46] that evaluated the effect of Pilates exercises on improving balance in older adults, Pilates was pointed out to be a helpful strategy for enhancing older adults’ balance, which may lower their chance of falling. Hence, fall prevention is paramount, as falls can lead to severe injuries and a decline in overall quality of life. Moreover, balance, encompassing both static and dynamic components, plays a pivotal role in preserving cognitive functions. By subjecting seniors to balance assessments before and after intervention programs, we can gauge the effectiveness of these strategies in averting falls and cognitive impairments, as pointed out by the author Carrasco-Poyatos et al. [24].

Within the studies included in our systematic review, a discernible trend emerges, indicating a greater prevalence of women engaging in Pilates compared to men. This observed discrepancy in the participants’ sex prompts contemplation on the potential factors influencing Pilates participation. Societal perceptions, marketing strategies, and the inherent nature of Pilates exercises may collectively contribute to this observed trend [49].

Therefore, physical exercise based on resistance or Pilates training performed in moderate-to-high workloads will interact positively within the mental health status of older women, helping to maintain their autonomy and daily life tasks [24]. These studies have highlighted the interconnectedness of balance with various components of physical and psychological well-being, reinforcing the significance of Pilates interventions to enhance balance in older adults. Therefore, Pilates training may offer a safe, adaptative, and effective means of improving balance and reducing the risk of falls, ultimately contributing to improved quality of life for this population.

### Strengths, Limitations, and Perspectives

Based on the findings of this review, there is potential for Pilates training to be incorporated into fall-prevention programs for older adults. Healthcare professionals and fitness instructors should consider older adult’s individualized needs and preferences when designing Pilates interventions. Adherence to Pilates programs may be enhanced by tailoring exercises to address specific balance deficits and ensuring that training is safe and enjoyable for participants.

This systematic review and meta-analysis is not without limitations. The potential for publication bias, heterogeneity among studies, and the presence of methodological flaws in some included studies must be acknowledged. These limitations may have influenced the pooled and isolated results and should be considered when interpreting the findings. The aggregated results from the reviewed studies revealed a mixed picture regarding the impact of Pilates training on balance in older adults. The observed variability in study outcomes can be attributed to several factors. First, methodological differences among the included studies may have influenced the results. Variations in the duration and intensity of Pilates training, the specific exercises employed, and the duration of follow-up could account for the disparities in findings. This point should be considered by fitness instructors who can develop progressive intensity and volume in Pilates interventions, aiming to potentialize functional gains. Furthermore, differences in the characteristics of the study populations, such as baseline balance levels and age distribution, may have played a role in the varied outcomes. These variations in measurement instruments underscore the multifaceted approach to static balance assessment within the context of Pilates training interventions. The same principle is applicable to the evaluation of dynamic balance, as elucidated in the Section 3, even though the large positive effects on balance reported in the meta-analysis were consistent across all of the included studies.

Future research in this field should address the methodological variations among studies, including the standardization of Pilates interventions and the incorporation of larger sample sizes and longer follow-up periods. Investigating the underlying mechanisms responsible for the observed balance improvements will provide valuable insights into the physiological and biomechanical effects of Pilates training in older adults.

## 5. Conclusions

In conclusion, this systematic review and meta-analysis underscores the potential of Pilates training as a valuable intervention to enhance balance in the elderly population. These findings contribute to the body of evidence regarding Pilates training and the incorporation of interventional programs into geriatric health and wellness initiatives, aiming to mitigate fall-related risks and improve older adults’ functionality and quality of life.

As healthcare strategies continue to evolve, incorporating Pilates into geriatric health initiatives emerges as a promising avenue to not only address the multifaceted challenges associated with aging but also to elevate the quality of life among older individuals. This study advocates for the thoughtful integration of Pilates into geriatric care, emphasizing its potential to serve as a cornerstone in fostering a healthier and more active lifestyle for the aging population.

## Figures and Tables

**Figure 1 healthcare-11-03083-f001:**
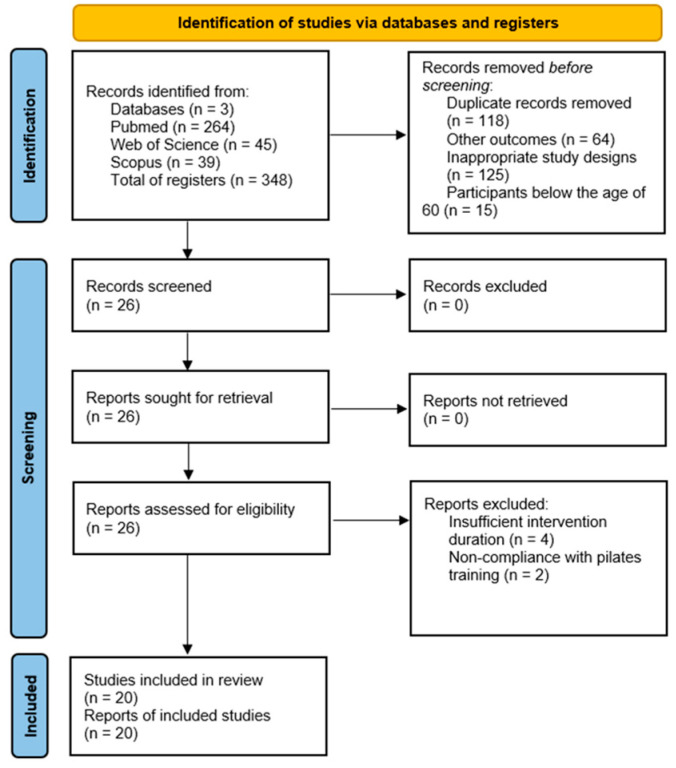
Flowchart of the systematic literature review.

**Figure 2 healthcare-11-03083-f002:**
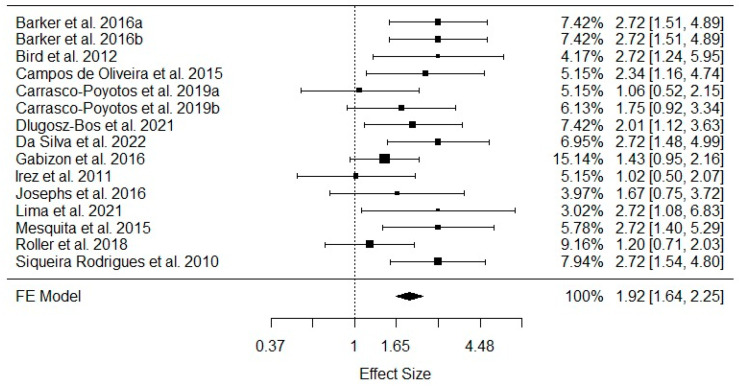
Pooled meta-analysis enrolling dynamic and statistic balance effect sizes in older adult Pilates practitioners [21,23,25,26,27,28,29,30,31,32,33,34,35]. FE: fixed-effects model; a: static balance outputs; b: dynamic balance outputs.

**Figure 3 healthcare-11-03083-f003:**
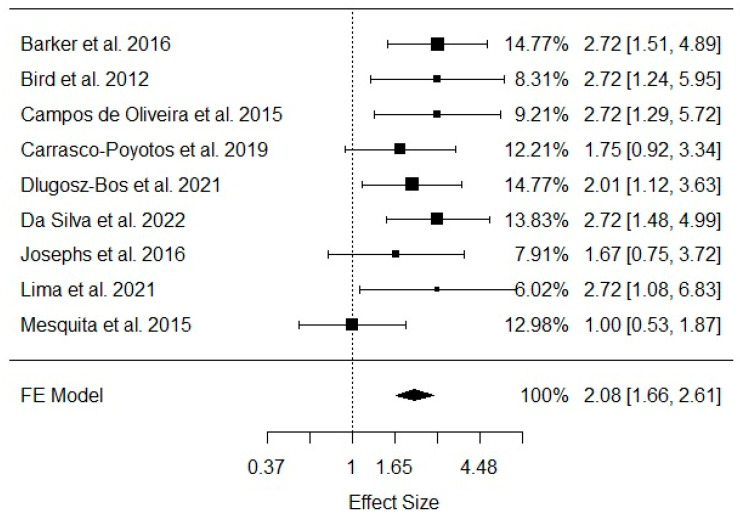
Pooled meta-analysis enrolling dynamic balance effect sizes in older adult Pilates practitioners [21,23,25,26,27,28,29,32,33]. FE: fixed-effects model.

**Figure 4 healthcare-11-03083-f004:**
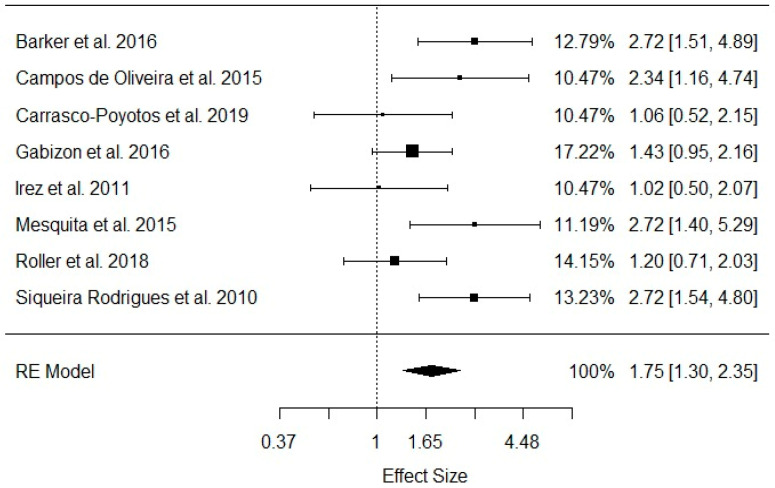
Pooled meta-analysis enrolling static balance effect sizes in older adult Pilates practitioners [23,25,26,27,30,31,34,35]. RE: random-effects model.

**Table 1 healthcare-11-03083-t001:** Study characteristics of the included studies for the meta-analysis.

First Author (Publication Year)	Sample Size	Age (M ± SD)	Sex	Volume and Frequency	Intervention Duration (Weeks)
Barker et al. (2016) [25]	IG: 20	IG: 69.25 ± 6.74	Both	60 min, 2 times/week	12
	CG: 29	CG: 69.41 ± 5.76			
Bird et al. (2012) [21]	T:27	T: 67.3 ± 6.5	Both	2 times/week	6
Campos de Oliveira et al. (2015) [26]	IG: 16	IG: 63.62 ± 1.02	Female	60 min, 2 times/week	12
	CG: 16	CG: 64.21 ± 0.80			
Carrasco-Poyotos et al. (2019) [27]	IG: 20 CG: 20	IG: 67.5 ± 3.87 CG: 65.89 ± 4.54	Female	60 min, 2 times/week	18
Długosz-Boś et al. (2021) [28]	IG: 30 CG: 20	IG: 67.73 ± 4.10 CG: 68.10 ± 3.35	Both	45 min, 2 times/week	12
Da Silva et al. (2022) [29]	IG: 29 CG: 32	IG: 68.82 ± 3.98 CG: 70.65 ± 6.06	Both	2 times/week	12
Gabizon et al. (2016) [30]	IG: 44 CG: 44	IG: 70.3 ± 3.8 CG: 72.1 ± 4.6	Both	3 times/week	12
Irez et al. (2011) [31]	IG: 30 CG: 30	IG: 72.8 ± 6.7 CG: 78.0 ± 5.7	Both	60 min, 3 times/week	12
Josephs et al. (2016) [32]	IG: 13 CG: 11	IG: 75.6 ± 6.2 CG: 74.5 ± 6.9	Both	60 min, 2 times/week	12
Lima et al. (2021) [33]	IG: 10 CG: 10	IG: 76.5 ± 5.93 CG: 75.8 ± 4.44	Both	60 min, 2 times/week	8
Mesquita et al. (2015) [23]	IG: 20 CG: 20	IG: 67.3 ± 4.9 CG: 71.5 ± 6.2	Female	3 times/week	4
Roller et al. (2018) [34]	IG: 27 CG: 28	IG: 78.52 ± 7.57 CG: 76.68 ± 6.79	Both	45 min, 1 time/week	10
Siqueira Rodrigues et al. (2010) [35]	IG: 27	T: 66.0 ± 4.0	Female	60 min, 2 times/week	12
	CG: 25				

Note: M: mean; SD: standard deviation; IG: intervention group; CG: control group; T: total.

**Table 2 healthcare-11-03083-t002:** Methodological quality and risk of bias in individual studies.

First Author (Publication Year)	Item 1	Item 2	Item 3	Item 4	Item 5	Item 6	Item 7	Item 8	Item 9	Item 10	Item 11	Item 12	Item 13	Item 14
Aibar-Almazán et al. (2019) [22]														
Barker et al. (2016) [25]														
Bird et al. (2012) [21]														
Campos de Oliveira et al. (2015) [26]														
Carrasco-Poyotos et al. (2019) [27]														
Carrasco-Poyotos et al. (2019) [24]														
Curi et al. (2018) [38]				NR	NR									
Długosz-Boś et al. (2021) [28]				NR										
Donath et al. (2016) [39]														
Da Silva et al. (2022) [29]														
Gabizon et al. (2016) [30]														
Irez et al. (2011) [31]				NR	NR									
Irez (2014) [36]		NR	NR	NR	NR									
Josephs et al. (2016) [32]														
Lima et al. (2021) [33]				NR	NR									
Mesquita et al. (2015) [23]														
Roller et al. (2018) [34]														
Siqueira Rodrigues et al. (2010) [35]				NR	NR		NR	NR						
Sofianidis et al. (2017) [37]				NR	NR									
Vieira et al. (2017) [40]				NR										

Note: dark gray: Yes; light gray: No; NR: not reported.

**Table 3 healthcare-11-03083-t003:** Summary of static balance measurement results.

Author (Publication Year)	Balance Measurement Instruments	Main Results
Aibar-Alzamán et al. (2019) [22]	Force platform	The participants in the Pilates group demonstrated statistically significant improvements in both the speed and anteroposterior movements of their center of pressure (COP), with eyes open and closed.
Barker et al. (2016) [25]	Force platform	Significant improvements in the Pilates group in the timed stance on foam with eyes closed.
Bird et al. (2012) [21]	Force platform	Static balance significantly improved during the study and from pre- to post-Pilates.
Carrasco-Poyatos et al. (2019) [27]	Force platform	The Pilates group did not exhibit a significant decrease in the displacement amplitude of the COP in the anteroposterior plane during the two-leg static balance test.
Carrasco-Poyatos et al. (2019) [24]	Force platform	There are no differences between groups regarding static balance.
Da silva et al. (2022) [29]	Force platform	This study improved postural stability by decreasing AP-ML sway for single-leg standing with eyes open, which results in fewer oscillations in the center of mass and improved sensory-motor function.
Długosz-Boś et al. (2021) [28]	Baropodometric platform	Significantly decreased values of the ellipse surface and mean values of velocity for the right foot were observed. The Limits of Stability test and the Modified Clinical Test of Sensory Interaction on Balance performed on an unstable surface with eyes closed indicated statistically significant changes.
Donath et al. (2016) [39]	Force platform	Moderate between-group effects were found following ANCOVA analyses for left-sided and right-sided postural sway during SLEO. Pairwise between-group effects were found in favor of BAL when compared with PIL only for right-sided SLEO. Multiple pairwise post-hoc *t*-tests revealed significant differences between pre-and post-testing for all groups during left-sided SLEO.
Gabizon et al. (2016) [30]	Force platform	No significant group-by-time interactions for any measure of balance.
Irez et al. (2011) [31]	Force platform	No significant differences were found between the Pilates and control groups in any of the sway measures.
Siqueira Rodrigues et al. (2010) [35]	Tinetti scale	Static balance significantly improved during the study and from pre- to post-pilates for the Pilates group.
Sofiadinis et al. (2017) [37]	Force platform	There was a significant reduction in trunk sway amplitude during the tandem stance with eyes closed and a decrease in COP displacement during the one-leg stance, indicating enhanced static balance.

Note: COP: center of pressure; AP-ML: anteroposterior and mediolateral sway; SLEO: single leg stance.

**Table 4 healthcare-11-03083-t004:** Summary of dynamic balance measurement results.

Author (Publication Year)	Balance Measurement Instruments	Main Results
Barker et al. (2016) [25]	TUG Dynamic gait index	There were significant improvements in the intervention group compared with the control group for the TUG test and dynamic gait index.
Bird et al. (2012) [21]	FSST TUG	Dynamic balance significantly improved during the study and from pre- to post-Pilates.
Campos de Oliveira et al. (2015) [26]	TUG Berg balance scale	In the intra-group comparisons of the evaluations, significant differences were found in the Berg balance scale and TUG measurements. Only a significant inter-group difference at post-intervention was found in the TUG measurements.
Carrasco-Poyatos et al. (2019) [27]	TUG	There were no differences between groups regarding dynamic balance. Intra-group analysis showed TUG test results improved significantly.
Curi et al. (2018) [38]	Battery test	In the dynamic balance test, there was found a time effect, a group x time interaction, and a group effect. Following the intervention, this test revealed substantial differences across groups.
Da silva et al. (2022) [29]	TUG	There was a statistically significant time effect on the pre- and post-test TUG scores. There was no statistically significant difference between the groups, interaction between groups, and time.
Długosz-Boś et al. (2021) [28]	TUG	The TUG test did not indicate any statistically significant differences in the experimental group following the Pilates program.
Donath et al. (2016) [39]	Y-balance	Post-hoc testing merely revealed significant differences between pre- and post-testing for left- and right-sided tests.
Gabizon et al. (2016) [30]	Berg balance scale	There were significant effects of time on the Berg balance score.
Irez et al. (2011) [31]	Dynamic stability measurement platform	Dynamic balance improved in the intervention group when compared to the control group.
Irez et al. (2014) [36]	Berg balance scale	Statistically significant differences were found in the pre- and post-intervention scores for balance.
Josephs et al. (2016) [32]	TUG Fullerton advanced balance scale	There was significant improvement in the Fullerton advanced balance scale for the intervention group.
Lima et al. (2021) [33]	TUG	The duration of the TUG test decreased significantly in the Pilates group from baseline to post-intervention moments.
Mesquita et al. (2015) [23]	TUG Berg balance scale Functional reach test	Significant differences were found between the control and Pilates groups in the TUG and functional reach test. PG exhibited improved performance in the TUG test and functional reach test compared with women in the CG. In a within-group comparison, women in the PG showed significant improvements in the functional reach test, TUG test, and Berg balance scale scores.
Roller et al. (2018) [34]	TUG	There was a statistically significant interaction between the intervention and time on TUG scores. Subjects in the Pilates group significantly decreased their TUG scores over time, whereas there was no significant change in TUG scores within the control group over time.
Sofiadinis et al. (2017) [37]	Force platform	An increase in trunk oscillation amplitude was observed during the sway task, reflecting improved dynamic balance.
Vieira et al. (2017) [40]	TUG	An improvement in dynamic balance was evident through a decrease in completion time during the TUG test.

Note: TUG: timed up and go test; FSST: four square step test; CG: control group.

**Table 5 healthcare-11-03083-t005:** Sensitivity analysis for the pooled meta-analysis between dynamic and static balance effect sizes.

First Author (Publication Year)	Estimate	SE	Z	*p*	CI Lower	CI Upper	Q	Qp	*I* ^2^	H^2^
Barker et al. (2016a) [25]	0.62	0.08	7.35	0.00	0.46	0.79	17.60	0.17	26.12	1.35
Barker et al. (2016b) [25]	0.62	0.08	7.35	0.00	0.46	0.79	17.60	0.17	26.12	1.35
Bird et al. (2012) [21]	0.64	0.08	7.64	0.00	0.47	0.80	18.26	0.15	28.80	1.40
Campos de Oliveira et al. (2015) [26]	0.66	0.08	7.81	0.00	0.49	0.82	18.96	0.12	31.44	1.46
Carrasco-Poyotos et al. (2019a) [27]	0.64	0.08	7.65	0.00	0.48	0.81	18.73	0.13	30.59	1.44
Carrasco-Poyotos et al. (2019b) [27]	0.68	0.08	8.16	0.00	0.52	0.85	16.19	0.24	19.70	1.25
Dlugosz-Bos et al. (2021) [28]	0.65	0.08	7.64	0.00	0.48	0.82	19.02	0.12	31.65	1.46
Da Silva et al. (2022) [29]	0.63	0.08	7.40	0.00	0.46	0.79	17.70	0.17	26.54	1.36
Gabizon et al. (2016) [30]	0.70	0.09	7.94	0.00	0.53	0.88	16.76	0.21	22.43	1.29
Irez et al. (2011) [31]	0.69	0.08	8.19	0.00	0.52	0.85	15.79	0.26	17.67	1.21
Josephs et al. (2016) [32]	0.66	0.08	7.90	0.00	0.50	0.82	18.92	0.13	31.29	1.46
Lima et al. (2021) [33]	0.64	0.08	7.73	0.00	0.48	0.80	18.48	0.14	29.66	1.42
Mesquita et al. (2015) [23]	0.63	0.08	7.50	0.00	0.47	0.80	17.94	0.16	27.53	1.38
Roller et al. (2018) [34]	0.70	0.09	8.17	0.00	0.53	0.87	15.67	0.27	17.06	1.21
Siqueira Rodrigues et al. (2010) [35]	0.62	0.09	7.31	0.00	0.46	0.79	17.49	0.18	25.66	1.35

Note: SE: standard error; Z: Z-statistics; *p*: level of significance of 95%; Q: Cochran statistics; Qp: *p*-value for the Cochran statistics; *I*^2^: test for between-study heterogeneity; H^2^: estimated heterogeneity; a: static balance outputs; b: dynamic balance outputs.

**Table 6 healthcare-11-03083-t006:** Sensitivity analysis for the meta-analysis of dynamic balance effect sizes.

First Author (Publication Year)	Estimate	SE	Z	*p*	CI Lower	CI Upper	Q	Qp	*I* ^2^	H^2^
Barker et al. (2016) [25]	0.69	0.12	5.50	0.00	0.44	0.93	7.70	0.36	9.08	1.10
Bird et al. (2012) [21]	0.71	0.12	5.89	0.00	0.47	0.95	8.14	0.32	14.02	1.16
Campos de Oliveira et al. (2015) [26]	0.71	0.12	5.84	0.00	0.47	0.94	8.08	0.33	13.40	1.15
Carrasco-Poyotos et al. (2019) [27]	0.76	0.12	6.16	0.00	0.52	1.00	8.31	0.31	15.77	1.19
Dlugosz-Bos et al. (2021) [28]	0.74	0.12	5.92	0.00	0.49	0.98	8.61	0.28	18.71	1.23
Da Silva et al. (2022) [29]	0.69	0.12	5.56	0.00	0.45	0.93	7.77	0.35	9.88	1.11
Josephs et al. (2016) [32]	0.75	0.12	6.26	0.00	0.52	0.99	8.30	0.31	15.69	1.19
Lima et al. (2021) [33]	0.72	0.12	6.02	0.00	0.48	0.95	8.28	0.31	15.49	1.18
Mesquita et al. (2015) [23]	0.84	0.12	6.82	0.00	0.60	1.09	2.59	0.92	0.00	0.37

Note: SE: standard error; Z: Z-statistics; *p*: level of significance of 95%; Q: Cochran statistics; Qp: *p*-value for the Cochran statistics; *I*^2^: test for between-study heterogeneity; H^2^: estimated heterogeneity.

**Table 7 healthcare-11-03083-t007:** Sensitivity analysis for the meta-analysis of static balance effect sizes.

First Author (Publication Year)	Estimate	SE	Z	*p*	CI Lower	CI Upper	Q	Qp	*I* ^2^	H^2^
Barker et al. (2016) [25]	0.49	0.16	3.13	0.00	0.18	0.80	11.17	0.08	46.43	1.86
Campos de Oliveira et al. (2015) [26]	0.52	0.17	3.14	0.00	0.20	0.85	13.02	0.04	54.68	2.20
Carrasco-Poyotos et al. (2019) [27]	0.61	0.16	3.89	0.00	0.31	0.92	11.85	0.07	49.59	1.98
Gabizon et al. (2016) [30]	0.60	0.18	3.36	0.00	0.25	0.95	12.81	0.05	53.02	2.12
Irez et al. (2011) [31]	0.62	0.16	3.97	0.00	0.31	0.92	11.52	0.07	48.11	1.92
Mesquita et al. (2015) [23]	0.50	0.16	3.15	0.00	0.19	0.81	11.81	0.07	49.70	1.98
Roller et al. (2018) [34]	0.62	0.16	3.80	0.00	0.30	0.94	11.70	0.07	48.49	1.94
Siqueira Rodrigues et al. (2010) [35]	0.49	0.16	3.13	0.00	0.18	0.79	10.95	0.09	45.24	1.82

Note: SE: standard error; Z: Z-statistics; *p*: level of significance of 95%; Q: Cochran statistics; Qp: *p*-value for the Cochran statistics; *I*^2^: test for between-study heterogeneity; H^2^: estimated heterogeneity.

## Data Availability

The data are not publicly available for privacy reasons.

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
