# Peer review of "The Effectiveness of Pilates Training Interventions on Older Adults’ Balance: A Systematic Review and Meta-Analysis of Randomized Controlled Trials"

_healthcare, 2023, doi:10.3390/healthcare11233083_

Round 1

Reviewer 1 Report

Comments and Suggestions for Authors

Dear authors Sampaio and colleagues,

I am grateful for the opportunity to participate in the peer review of the article titled "The Effectiveness of Pilates Training Interventions on Older Adults' Balance: A Systematic Review and Meta-Analysis of Randomized Controlled Trials." It is a pleasure to contribute to the review process of this work.

In general, I find the article to be well-presented and well-founded. I appreciate the rigor and systematic approach you have taken in your research. However, I would like to suggest an additional consideration to further enrich the study.

It would be beneficial if the authors could consider conducting a meta-regression. My concern arises from the observation that certain variables, such as the gender composition of participants (some studies included only women, while others included both men and women) and differences in balance measurement instruments, might be influencing the outcomes. I believe that a meta-regression could offer a deeper and more detailed insight, helping to determine whether these variables significantly impact the results of the meta-analysis.

This suggestion is proposed with the intention of strengthening the analysis and providing a more nuanced understanding of the effects of Pilates training interventions on the balance of older adults.

I again thank you for the opportunity to review this work. Your study's contribution to the field is noteworthy, and I look forward to seeing your research published.

Sincerely,

Reviewer 2 Report

Comments and Suggestions for Authors

First of all, congratulations on the work done, then I will mention a number of changes and recommendations in order to obtain clearer and more accurate information.

Revise the title.

- Comments on the abstract:

Line 31: I think you should write “adults’ balance” instead of “adult’s balance”.

Line 33: the same with participant’s.

Revise this type of expression in all the manuscript.

- Comments on the introduction:

Lines 48-50. What do you mean talking about distance between the lower limbs?

- Comments on results:

Check grammar and fluency in all table 4, as an example, “Individuals in the Pilates group improved statistically significantly in velocity…”, this phrase sounds very bad and is not very understandable, check all similar expressions.

Line 268. You do not analyze 15 studies, you have analyzed 13 studies, but you included 15 groups of training or outcomes.

In general, the results are understandable, but I am missing a more graphic tool to evaluate the studies, such as Cochrane's RoB2.

Check the fluency and sense of expressions throughout the manuscript to improve readability.

Reviewer 3 Report

Comments and Suggestions for Authors

Line 64-66

Exercises that combine body and mind are used and require stability of the trunk, strength, and flexibility, as well as a concentration on body control, posture, and breathing.

Coment 1

Explain a little combine body and mind“, this statement needs a reference.

Line 66-67

Six key elements are used: center, concentration, control, precision, fluidity, and breathing.

Coment 2

What do you think are the key elements, please explain? Be a little more precise because, for example, breathing is common during every exercise.

Table 1. Items of the NIH Quality Assessment Tool for Controlled Intervention Studies.

Coment 3

I see no reason why you should display an empty table. In the text, it is sufficient to report that you used the National Institutes of Health (NIH) recommendations, and the table mentioned can serve as supplementary material.

Line 175

Finally, 20 studies that met the criteria and aims of this systematic review were in- 175

cluded, as shown in Figure 1.

Table 2. Study Characteristics of the Included Studies for the Meta-analysis.

Coment 4

20 studies met criteria, only 13 in table 2. I did not see where it was explained why some studies did not enter the meta-analises.

Coment 5

It would be good to indicate in the table that the duration of treatment is given in weeks. There is no need to search for that information in text.

Table 4. Cont.

Coment 6

Add full table title.

Line 374-377

These results align with previous research findings, as a prior systematic review conducted by Bullo et al. [40] observed that pilates interventions yield various favorable outcomes, including improvements in balance, reduced risk of falling, increased functional mobility, and enhanced postural stability.

Coment 7

Balance, mobility and stability are measurable categories. Risk is not a directly measurable category. We can say that with increasing stability, the risk of falling decreases. However, subjects with better stability are less careful. The risk of falling should be mentioned more carefully.

Line 467-471

Future research in this field should address the methodological variations among studies, including the standardization of Pilates interventions and the incorporation of larger sample sizes and longer follow-up periods. Investigating the underlying mechanisms responsible for the observed balance improvements will provide valuable insights into the physiological and biomechanical effects of Pilates training in older adults.

Coment 8

This study confirms that Pilates exercises have a positive effect on stability. This is not a new finding, but this research apostrophizes it. We do not know if Pilates exercises are better than other exercises for improving stability? The point is not to better confirm the link between Pilates and stability, but to determine which exercises are most effective.

Coment 9

Conclusion is too short.

Coment 10

Women or both sexes predominate in the papers in the sample, I have not seen any papers on an exclusively male population. If women are more inclined to this exercise, that should be commented on somewhere.
